# Data-driven decentralized breeding increases prediction accuracy in a challenging crop production environment

Kauê de Sousa [1,2], Jacob van Etten[2], Jesse Poland [3], Carlo Fadda [4], Jean-Luc Jannink [5,6], Yosef Gebrehawaryat Kidane [4,7], Basazen Fantahun Lakew[4,8], Dejene Kassahun Mengistu[4,7], Mario Enrico Pè[7], Svein Øivind Solberg[1] & Matteo Dell'Acqua [7 ✉]

Crop breeding must embrace the broad diversity of smallholder agricultural systems to ensure food security to the hundreds of millions of people living in challenging production environments. This need can be addressed by combining genomics, farmers' knowledge, and environmental analysis into a data-driven decentralized approach (3D-breeding). We tested this idea as a proof-of-concept by comparing a durum wheat (*Triticum durum* Desf.) decentralized trial distributed as incomplete blocks in 1,165 farmer-managed fields across the Ethiopian highlands with a benchmark representing genomic prediction applied to conventional breeding. We found that 3D-breeding could double the prediction accuracy of the benchmark. 3D-breeding could identify genotypes with enhanced local adaptation providing superior productive performance across seasons. We propose this decentralized approach to leverage the diversity in farmer fields and complement conventional plant breeding to enhance local adaptation in challenging crop production environments.

[1] Department of Agricultural Sciences, Inland Norway University of Applied Sciences, Hamar, Norway. [2] Digital Inclusion, Bioversity International, Montpellier, France. [3] Department of Plant Pathology, Kansas State University, Manhattan, KS, USA. [4] Biodiversity for Food and Agriculture, Bioversity International, Nairobi, Kenya. [5] College of Agriculture and Life Sciences, Cornell University, Ithaca, NY, USA. [6] Agricultural Research Service, United States Department of Agriculture, Ithaca, NY, USA. [7] Institute of Life Sciences, Scuola Superiore Sant'Anna, Pisa, Italy. [8] Ethiopian Biodiversity Institute, Addis Ababa, Ethiopia. ✉email: m.dellacqua@santannapisa.it

The big data revolution in genomics has transformed plant breeding with inexpensive sequencing methods, enabling greatly accelerated variety development[1–3]. At present, plant breeders use data-driven methods, including genomic prediction, to increase selection intensity while reducing the time of the breeding cycle and deriving greater genetic gain[4]. Most conventional breeding programs still rely on a centralized scheme aimed at maximizing genetic diversity (G) in the early stages of selection and then identifying superior germplasm based on phenotypic observations made in a limited number of research stations with explicit environmental (E) and management (M) conditions. In this setting, genomic prediction may be used to predict the performance of untested new genotypes but is bound to the G × E × M interactions captured by the research stations that are used to train the selection models[5]. This limitation of centralized breeding approaches may result in suboptimal development and deployment of crop varieties for use by farmers seeking local adaptation in challenging environments[6]. This is especially relevant in smallholder farming systems, which involve about 80% of the world farmers[7] and call for tailored solutions to support food security.

To respond to local cropping needs impacted by climate change, breeders need to find new ways to accelerate variety development while directly addressing G × E × M interactions to the fullest[3,8,9]. Mobilizing farmers' traditional knowledge of crop varieties and local adaptation can address this challenge and enhance adoption of improved varieties[6,10–12] in a coherent, decentralized breeding program relying on farmer-participatory selection[13–15]. A crowdsourced citizen science approach has demonstrated the feasibility of a data-driven decentralized variety evaluation[16] that enables on-farm variety testing in a digitally supported and cost-efficient way[17]. Predictive accuracy of farmer selection criteria may outperform breeder evaluations even in a context of modern agriculture[18].

Crowdsourced citizen science further integrates the E and M components into breeding by performing selection directly in target environments and using environmental data to analyze genotypic responses. Thus, the citizen science approach scales E and M data collection to generate a volume of data that matches the big data dimension of G. Combining genomic prediction with citizen science opens the possibility of simultaneously capturing the three dimensions of crop performance, G, E, and M, in a data-driven way. Here, we describe and demonstrate potential benefits of this approach that we call *data-driven decentralized breeding*, or 3D-breeding, for short. Potentially, 3D-breeding could benefit the ~500 million smallholder farmers around the world who often produce in challenging, low-input environments and work with diverse cropping and farming systems and respond to local consumption preferences[7].

We applied the 3D-breeding approach in the Ethiopian highlands, where many smallholder farmers grow durum wheat (*Triticum durum* Desf.) and select landraces following criteria related to environmental adaptation, food culture, and market demand[19,20]. Rich local wheat diversity has co-evolved with local cultures and landscapes over millennia. Consequently, Ethiopian farmers still often select and cultivate local landraces, which under local conditions tend to outperform modern varieties produced by centralized breeding[21]. In this context, 3D-breeding can leverage local wheat diversity and knowledge and bring breeding closer to the target environments cutting through the complexity of G × E × M.

Here, we collected data from the genotyping and phenotyping of 400 wheat varieties in centralized stations commonly used for varietal selection in Ethiopian highlands. We then selected and distributed a subset of 41 genotypes as packaged sets containing incomplete blocks of three genotypes, plus one commercial variety to each of 1,165 farmers located in the same breeding mega-environment. We tested 3D-breeding against a competitive benchmark that represents breeding based on a genomic prediction model trained on centralized stations to predict varietal performance in farmers' decentralized fields. We focused on grain yield (GY) and farmers' overall appreciation (OA) of wheat genotypes, which were both recorded in centralized and decentralized trials. To establish the benchmark, we used a genomic prediction model trained on data measured in stations to predict wheat GY and OA in farmer fields (Fig. 1a). We then developed 3D-breeding to move the selection to farmer fields, predicting wheat performance in farmers' fields using a decentralized approach (Fig. 1b). Comparing side by side the accuracy of the two methods, we found that that 3D-breeding could increase prediction accuracy in challenging environments and thus complement genomics assisted breeding.

## Results and discussion

**Performance of centralized breeding based on genomic prediction and farmers' traditional knowledge.** Heritability ($H^2$), the proportion of phenotypic variance explained by genotypic variance, was 0.55 and 0.42 for $GY_{STATION}$ across locations for 2012 and 2013 respectively (Supplementary Data 1). To capture farmers' traditional knowledge regardless of gender, farmer scores were combined across men and women respondents: the $H^2$ of $OA_{STATION}$ was 0.78 across locations. Narrow sense heritability ($h^2$) was calculated considering genetic co-variance of genotypes and provided more conservative estimates for all traits, yet $OA_{STATION}$ was consistently more heritable than $GY_{STATION}$ (Supplementary Data 1 and 2). We validated the centralized benchmark by predicting on-station performance from one season to the next, focusing on a subset of 41 genotypes that were later distributed in decentralized farmer fields. This led to accuracies up to $\tau = 0.248$ in predicting $GY_{STATION}$ in the following season (Supplementary Fig. 1). Previous studies showed that men and women may prioritize different traits depending on their role in the farming activity, from cropping to marketing of products[22,23]. In our study, gender differences in $OA_{STATION}$ scoring are reflected by different $H^2$ achieved by men (0.84) and women (0.67), with a more marked difference in Hagreselam (Supplementary Data 2). Still, men and women provided consistent evaluations (Supplementary Fig. 2). This is in line with tricot observations reporting that gender have low overall effect on varietal choice[17] and shows that farmer scores are reliable measures of genotypes performance. Indeed, we found that $OA_{STATION}$ was a better predictor than $GY_{STATION}$ to capture both $OA_{STATION}$ and $GY_{STATION}$, including when disaggregated by gender (Supplementary Fig. 3). Previous studies explored the relation between OA and agronomic performance of wheat, showing that farmers' appreciation was positively correlated to yield, seed size, biomass, and negatively correlated with time to flowering and time to maturity[20,21].

**Benchmark: using centralized measures to predict performance in farmer fields.** The benchmark had a low prediction accuracy when using $GY_{STATION}$ to predict $GY_{FARM}$ in individual seasons, with an average of $\tau = 0.046$. When using $OA_{STATION}$ to predict $OA_{FARM}$, the average was $\tau = 0.141$ (Table 1). Indeed, GY and OA collected in stations were poorly correlated with on-farm performance (Supplementary Fig. 4). Accuracy remained low when $GY_{STATION}$ was used to predict measures of $GY_{FARM}$ and $OA_{FARM}$ combined across seasons and in alternative scenarios considering different subsets of training and test populations (Supplementary Data 3). Interestingly, $OA_{STATION}$ had consistent positive accuracy in predicting $GY_{FARM}$ and $OA_{FARM}$ (Supplementary Fig. 5). This confirmed that genomic prediction can be

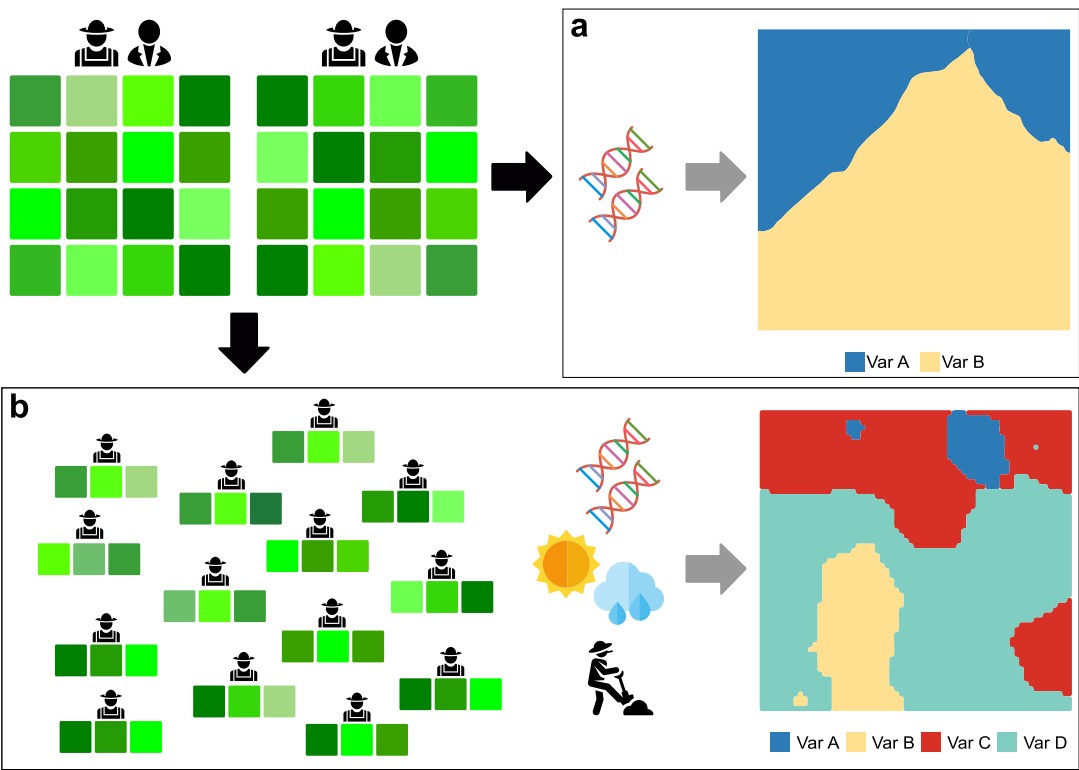

**Fig. 1 A comparison of centralized versus decentralized breeding approaches.** Centralized breeding (**a**) derives recommendations from breeders' evaluation and possibly participatory assessments in a limited set of stations, using genomics to accelerate the production of varieties that are eventually recommended with coarse spatial resolution. The plot shows the broad recommendation space of two hypothetical varieties, Var A and Var B. This system may become more efficient if complemented by 3D-breeding (**b**), a decentralized approach where the best candidate genotypes are tested by farmers in small, blinded and randomized sets. 3D-breeding produces scalable solutions that can be linked to genomics, farmers' knowledge and environmental data, to enhance the local adaptation of the resulting varieties and tailor their recommendation to the landscape. This is represented in the plot to the right by the precise recommendation space of hypothetical varieties Var A, Var B, Var C and Var D.

**Table 1 Performance of the 3D-breeding compared with the benchmark of a centralized genomic prediction.**

| Approach | OA | GY |
|---|---|---|
| **Centralized GS** | | |
| Season 1 ($n=179$) | 0.134 | −0.012 |
| Season 2 ($n=651$) | 0.105 | 0.076 |
| Season 3 ($n=335$) | 0.183 | 0.073 |
| | **0.141 (± 0.039)** | **0.046 (± 0.049)** |
| **3D-breeding** | | |
| Season 1 ($n=179$) | 0.270 | 0.160 |
| Season 2 ($n=651$) | 0.276 | 0.078 |
| Season 3 ($n=335$) | 0.203 | 0.119 |
| | **0.251 (± 0.040)** | **0.109 (± 0.041)** |

3D-breeding provides higher across-season goodness-of-fit (Kendall $\tau$) than centralized genomic prediction on overall appreciation (OA) and grain yield (GY) derived from farmer rankings on decentralized fields.
Prediction accuracy combined across seasons is given in bold.

enhanced by farmers' traditional knowledge whereas selection based only on GY could result in reduced appreciation by farmers (Supplementary Fig. 6).

$GY_{STATION}$ provided a more accurate prediction of $GY_{FARM}$ when restricting the model to cold-tolerant genotypes (Supplementary Fig. 7). This was likely due to the partial representation of the climatic variation that can be provided by a centralized approach with a handful of stations (Supplementary Fig. 8), as farms could experience lower temperatures than stations (Supplementary Fig. 9). Still, centralized predictions of increasingly distant farm environments shown an erratic pattern, showing that variation at the

farming sites goes beyond that captured by temperature variation (Supplementary Fig. 10). Regardless the fact that both stations and farms were located in the same agroecological zone (Supplementary Fig. 11), the benchmark failed to predict performance under production conditions, showing that the small-scale variation in climate and management may hamper the success of centralized breeding decisions.

**3D-breeding provides higher prediction accuracy than the benchmark.** Model predictions from 3D-breeding consistently provided higher accuracy than the benchmark for $GY_{FARM}$ and

$OA_{FARM}$ with $\tau = 0.109$ and $\tau = 0.251$ (Table 1). When supported by smaller sets of observations (from 5% to 75% of the available data), 3D-breeding maintained superior accuracy than the benchmark, with a mean accuracy spanning from $\tau = 0.162$ to $\tau = 0.230$ for $OA_{FARM}$ and from $\tau = 0.076$ to $\tau = 0.106$ for $GY_{FARM}$ (Supplementary Data 4). The prediction accuracy of the 3D-breeding approach was not biased towards specific environmental conditions, suggesting that it could capture the environmental diversity of test sites better than the benchmark (Supplementary Figs. 12 and 13).

Overall appreciation of genotypes in 3D-breeding provided higher prediction accuracies than $GY_{FARM}$ in all farmers' fields (Supplementary Fig. 14). Previous studies showed that farmer evaluations are able to capture agronomic performance of genotypes in untested locations[18,20], as confirmed by the high $H^2$ observed for $OA_{STATION}$ (Supplementary Data 2). Farmers provided OA according to their own experience and preferences, and it presumably depended on a combination of traits of which GY represented only one dimension[21]. By eliciting traditional knowledge of men and women farmers at cropping sites, 3D-breeding successfully predicted varietal performance under local growing conditions (Supplementary Fig. 5). $GY_{FARM}$ is objectively and independently measured at each plot and therefore it could not be biased by $OA_{FARM}$. It is possible that $GY_{STATION}$ and $GY_{FARM}$ failed to capture secondary traits with high heritability (Supplementary Data 1) that were observed by farmers and that were correlated to the $GY_{FARM}$ of genotypes under field conditions[20,21]. As $OA_{FARM}$ is directly related to the probability of variety adoption it is an important complement to GY in driving varietal development for challenging environments.

**Superior genotype selection with 3D-breeding is consistent across seasons.** We extrapolated the 3D-breeding model predictions to assess the probability that the genotypes selected by 3D-breeding based on OA will outperform currently recommended varieties[24]. We found that the best three genotypes in each terminal node of the 3D-breeding model splits had a genetic background markedly separated from that of varieties currently recommended for the region, and consistently higher *worth* (Fig. 2a). Indeed, the model selected genotypes derived from landraces over improved varieties. We estimated the reliability, i.e. the probability that the model recommendation exceeds the current recommendation in terms of $OA_{FARM}$. In this assessment, predictions from 3D-breeding outperformed the current varietal recommendations in most of the farmers' fields, with consistent high reliability (0.83–0.91), including in challenging areas for which the centralized breeding approach could not provide accurate predictions (Fig. 2b). To provide an agronomic measure, we also predicted the increase in $GY_{FARM}$ and tested to see if the yield advantage could be maintained by selecting the best three genotypes indicated by 3D-breeding under 15 different growing seasons simulated on target farms. We found that 3D-breeding ensured consistent recommendations over years with expected increases in yield of about 20% (Fig. 2c). Thus, 3D-breeding accurately identified the best performing genotypes to be advanced in breeding efforts targeting local growing conditions, to be developed into suitable new varieties, and to be promoted with environmental-specific recommendations.

**Implications for rethinking breeding programs.** Our results show that 3D-breeding is superior to a benchmark that represents a centralized breeding approach. The genomic prediction benchmark and 3D-breeding rely on different statistical designs and methods, yet they have the same aim: providing accurate prediction of phenotypes in untested environments. We believe that the implementation of the two approaches was realistic and

of high quality, making the comparison realistic. We have explored whether the superiority of 3D-breeding was sensitive to the influence of data availability, the geographical placing of the centralized selection environments or the variable of focus (overall appreciation or grain yield) and found that its superiority was robust. This has important implications for breeding program design.

Genomic prediction is a well-known approach to accelerate breeding programs, but current implementations in plant breeding have not yet been combined with a decentralized approach. The earliest and most successful implementations of genomic prediction have arguably occurred in dairy cattle breeding[25]. The accelerated evaluation of bull net merit was key to this[26], but that success also depended on the fact that breeders had access to phenotyping data from a broad range of environments in the form of milking records, which farmers record for their own management benefit. In conventional crop breeding, all of the phenotyping costs fall on the breeding program and limit the number of target environments that can be represented in the selection process. 3D-breeding seeks to complement and expand the flow of information from a few centralized locations to the whole mega-environment where results from numerous decentralized observations and farmer knowledge may converge to inform breeding decisions.

In centralized breeding, the environmental variation of target environments is factored through experimental control or indirectly as an average response across breeding stations as in our benchmark. This makes extrapolation to real farming conditions challenging. G × E affects yield and its components[27,28] and calls for selection models to explicitly account for it[29]. These models, however, are bound to the observations that can be made in resource-intensive breeding trials. The scope and size of the benchmark in this study was representative of a regional variety trial, an advanced stage in breeding focusing on a set of genetic materials and target environments with the aim of selecting the best genotypes for varietal release and recommendation. Even when they are place din relatively representative locations, centralized stations cannot represent the entire pedoclimatic space occupied by target farmer fields (Supplementary Fig. 9). Data from crowdsourced citizen science, like 3D-breeding, may further our understanding of the G × E interactions that are observed in farmer fields and allow the integration of increasingly accurate seasonal prediction models[30] in breeding and germplasm recommendation pipelines.

The 3D-breeding approach addresses the low correlation between performance in selection environments and production environments, while taking a step forward to fully data-driven breeding. In this, 3D-breeding is a promising approach that could add to conventional breeding increasing varietal performance in smallholder agriculture, which accounts for the largest share of the global farms[7]. In those settings, the adoption rate of current breeding innovation may be suboptimal due to socioeconomic and environmental factors[9,21,31–33]. Climate change is pushing these farming systems to the edge of their adaptation capacity with increasing pressure from pest and diseases[34,35], threats of yield loss[36,37] and increased seasonal climatic variability[38,39], calling for tailored solutions. 3D-breeding may speed up the turnover of varietal release to address these challenges. As farmers are at the center of the experimental design, varieties deriving from 3D-breeding are more likely to be adopted and suited to local cultivation[11,40], increasing the effectiveness of breeding efforts. Indeed, we found that farmers' OA was a better predictor than GY in predicting yield realized both in centralized and decentralized trials (Table 1). Likewise, varieties derived from landraces consistently outranked the performance of improved varieties (Fig. 2a) derived from centralized breeding[19]. Beyond varietal recommendations, 3D-breeding can direct the choice of parents to

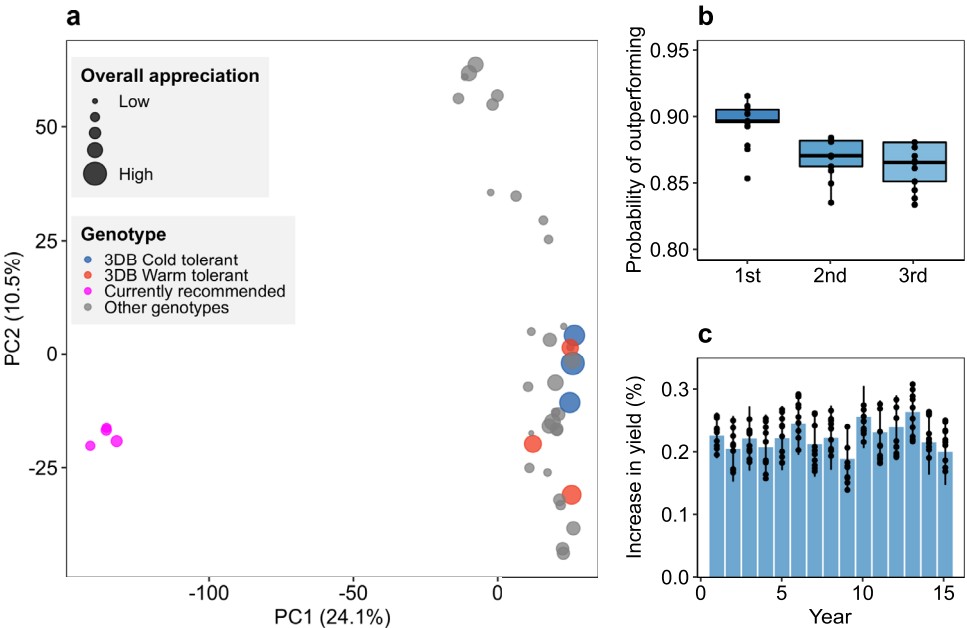

**Fig. 2 Selection of durum wheat (*Triticum durum* Desf.) genotypes based on 3D-breeding. a** Principal component coordinates of the genetic diversity of tested genotypes. Pink dots represent the varieties currently recommended for the area of study. 3DB Cold tolerant (blue) represents the top 3 genotypes selected by 3D-breeding in cold areas (minimum night temperature <11.5 °C). 3DB Warm tolerant (red) represents the top 3 genotypes selected by 3D-breeding in warm areas (minimum night temperature >11.5 °C). Size of dots represents the performance of genotypes in farmer fields as overall appreciation (OA). **b** Probability of outperforming improved varieties currently recommended by using genotype selection generated by 3D-breeding with OA. The panel shows the probability of the top 3 genotypes in a given location in outperforming the improved variety recommended for that location. **c** Expected increase in yield across 15 consecutive growing seasons (2001 to 2015) for genotype selection from 3D-breeding. $n = 1,165$ observations.

crosses aiming at the production of recombinant lines to provide higher and more stable yields in local agriculture.

**Potential of 3D-breeding for challenging cropping environments.** It has been advocated that scientific research and innovation must decidedly focus on small-scale farming systems to move towards a world with zero hunger by 2030[41]. 3D-breeding makes smallholder farmers innovation drivers as well as recipients, supporting the sustainable intensification of challenging environments. However, 3D-breeding is useful beyond small-holder farming agriculture, and the citizen science approach on which it relies has already been applied to several crops to enhance the selection of climate-adapted varieties[16]. Its general scheme may also be useful in high-input, yield maximizing agriculture to enhance local adaptation and support sustainability and food security, where the usefulness of farmers' evaluations in a genomic setting was already demonstrated[18]. In these settings, 3D-breeding could contribute to the identification and development of varieties with higher local adaptation, reducing the need of external inputs to achieve desired yields.

There are a number of open questions in relation to decentralized crop breeding, including how to best motivate new farmers to participate in the evaluation of materials, how much planting material each farmer needs, the logistics of providing farmers with the genetic material, and how to share benefits deriving from the utilization of farmers' knowledge to produce new varieties. Both in centralized stations and in decentralized fields, we found that farmers were eager to participate without material compensation. Farmers seek access to new genetic materials that they could not access otherwise, in exchange for the minimum investment of running small plots and providing a concise evaluation at the end of the season in the case of the tricot evaluation[17]. This happens even if some may not be adapted to their growing environment. Previous studies showed

that farmers perceive as beneficial the interaction with experts and the sharing of information[42]. Benefit to farmers may exceed the immediate access to improved technology, if the deeds to reconcile farmers' and breeders' rights in plant variety protection succeed[43].

In this study, farmers evaluated top performing varieties chosen from a larger set, but future studies may focus on larger collections of germplasm to be evaluated through 3D-breeding in combination with evaluations performed in research stations. These may include new genetic materials prioritized by speed-breeding[44] and haplotype-based selection[45]. Our results show that already the current replication level of the experimental design may support more diversity (Supplementary Data 4). 3D-breeding may be most effective as a complement to a centralized breeding system providing a high-throughput evaluation of correlated traits to support earlier varietal selection to be tested in farmer fields[46]. Our method may complement and enhance trait prioritization and speed-breeding methods currently used to reduce the need of extensive, resource-intensive multilocation trials[47]. Accuracy is just one among the factors controlling genetic gain[48], thus our findings should be integrated in the broader picture of modern breeding. Multi-trait models may increase prediction accuracy by measuring correlated traits with higher heritability[46,49,50]. These models could be employed in centralized stations and used to narrow down the set of varieties to be distributed to farmers in the 3D-breeding approach aiming to fine-tune local adaptation. Moreover, our findings support the need to further explore the challenge to model farmers' appreciation at the genomic level to improve the effectiveness of genotypes evaluation trials[18].

The advantages provided by the approach are clear: phenotyping costs would be divided in much smaller packets, supporting the modular expansion of the breeding effort towards new genetic materials or new locations. In return, each generated datapoint

would be a better representation of the true farming conditions to which varieties are directed. Previous research found that the involvement of farmers in selection experiments has negligible effects on costs[51]. In 3D-breeding the costs are shared by farmers, who would in exchange obtain access to the best materials for their farm. Farmer preference would be collected directly on farms rather than derived from correlated metrics that come from on-station evaluations in centralized breeding. In terms of absolute costs, an implementation of 3D-breeding based on OA would require additional investments in seed multiplication, seed distribution and telecommunications to obtain feedback from farmers. These costs are generally lower per data point than in on-farm evaluation trials using conventional approaches. Genotyping costs are negligible thanks to ever increasing sequencing capabilities[1].

## Conclusion
The data-driven focus of 3D-breeding enables embracing the complexity of real-world G × E for the benefit of breeding. Such a multidimensional, collaborative approach calls for best practices in data management and sharing[52]. 3D-breeding is based on a documented set of methods, from experimental design[17] to data curation and analysis[53,54]. While our demonstration of these methods relied on a large dataset, we believe that much larger field sample designs and genomic variant datasets are quite feasible and will provide additional power, as is also much in evidence in livestock genetics. The expansion of the design with the addition of further testing seasons and local management conditions may allow to highlight drivers of local performance of genotypes beyond temperature[55]. Further 3D-breeding studies may opt to stratify participants for socioeconomic features of interest, including gender, age, or income, to fully characterize traditional knowledge in its many dimensions. Ideally, 3D-breeding could be combined with conventional, centralized breeding to improve the training of prediction models to address local adaptation. Once new varieties are developed though the crowdsourced combination of breeders' and farmers' knowledge, future research shall focus on the potential impact of these methods on conservation and use of traditional agrobiodiversity both in situ and beyond the local environments in which it was developed. The crowdsourced citizen science approach associated with open-source digital tools makes it possible for breeders and farmers to apply 3D-breeding in new contexts and crops, dependent only on creativity in identifying untested production niches, potentiating a culturally driven co-evolution between farming systems and data-driven breeding to complement traditional breeding.

## Materials and methods
**Plant materials and DNA extraction**. We selected 400 durum wheat (*Triticum durum* Desf.) genotypes from a representative collection of landraces accessions maintained at the Ethiopian Biodiversity Institute (EBI) and improved lines cultivated in Ethiopia. Landrace accessions were purified to derive a uniform genetic background to undergo all subsequent analyses, so that all seeds derived from a single spike representative of the EBI accession as described in Mengistu et al. (2016)[19]. Genomic DNA was extracted from fresh leaves pooled from five seedlings for each of the purified accessions with the *GenElute*[TM] Plant Genomic DNA Miniprep Kit (Sigma-Aldrich, St Louis, USA) following manufacturer's instructions in the Molecular and Biotechnology Laboratory at Mekelle University, Tigray, Ethiopia. Genomic DNA was checked for quantity and quality by electrophoresis on 1% agarose gel and Nanodrop[TM] 2000 (Thermo Fisher Scientific Inc., Waltham, USA). Genotyping was performed on the Infinium 90k wheat chip at TraitGenetics GmbH (Gatersleben, Germany). Single nucleotide polymorphisms (SNPs) were called using the tetraploid wheat pipeline in GenomeStudio V11 (Illumina, Inc., San Diego, CA, USA). SNP calls were cleaned for quality by filtering positions and samples with failure rate above 80% and heterozygosity above 50%. Full details on the genotyping are given by Mengistu et al.[19]. The SNP calls for the genotypes included in this study and the details on the provenance of genotypes tested are given as part of the full dataset on Dataverse[56].

**Evaluation of genotypes in centralized trials**. Centralized trials were performed in 2012 and 2013 in the districts of Geregera (Amhara) and Hagreselam (Tigray) (Supplementary Fig. 15). The experimental stations were chosen to represent the highland agroecology of Ethiopia and are often used as varietal testing sites for local agriculture. The trial was laid out in a replicated alpha lattice design with the full set of 400 genotypes as entries, for a total of 800 plots in each field. Field managements were conducted as per local guidelines with manual weeding. Accessions were sown in four rows 2.5 m long, at a seeding rate of 100 kg ha$^{-1}$. At sowing, 100 kg ha$^{-1}$ diammonium phosphate and 50 kg ha$^{-1}$ urea were applied, with additional 50 kg ha$^{-1}$ urea at tillering.

In each location, 15 men and 15 women who were experienced smallholder farmers growing durum wheat were invited to evaluate plots during the 2012 season. After being informed on the study, its aims and methods, farmers provided a verbal informed consent that was recorded on paperwork. The evaluation was conducted at flowering time in each experimental station, for a total of 60 farmers involved. The farmers had no previous knowledge of the genotypes included in this study to prevent bias in the evaluations. The participants provided appraisal with Likert scales[57] to genotypes for overall appreciation (OA)[20,21], with 1 being worse and 5 the best. Prior the experiment, farmers were involved in focus group discussions and trained on how to perform the evaluation[21]. During the evaluation, farmers were divided in gender-homogenous groups of 5 people, were introduced in the field from random entry points, and were accompanied plot by plot by a researcher who guided the evaluation and collected OA values from individual farmers. Farmers did not use half-values to streamline the evaluation effort. After harvesting, technicians measured grain yield (GY) as grams of grain produced per plot, then converted into $t \cdot ha^{-1}$. Other agronomic traits were also collected as detailed in Mengistu et al. (2016)[19].

**Evaluation of genotypes in decentralized trials**. A total of 1,165 decentralized field, each with 4 plots, were established between 2013 and 2015 during three growing seasons across the regions of Amhara (471), Oromia (399) and Tigray (295) (Supplementary Fig. 15) using a subset of 38 purified landraces accessions identified through farmer evaluation in centralized trials[21] and three modern cultivars, for a total of 41 wheat genotypes (Supplementary Fig. 15). Farms were selected in areas representative for wheat growing in Ethiopia, based on previous history of cultivation of the crop (Supplementary Fig. 16). Individual farmers were engaged via local agricultural offices and selected based on their willingness to participate and of the following criteria: (i) being wheat growers, (ii) owning the land, (iii) living in the village all year. No financial incentive was given to farmers besides the opportunity to test new varieties and keep the harvest from the decentralized varietal plots. Farmers were fully informed of the study and provided a verbal informed consent that was recorded on paperwork. Selected farms were representative of the agroecological zones of the centralized fields (Supplementary Fig. 11). Season 1 (2013) comprised 179 fields, Season 2 (2014) comprised 651 fields, and Season 3 (2015) comprised 335 field. Differences in number of fields by season are due to availability of farmer communities. Trials (farmer-managed plots) followed the triadic comparison of technologies (*tricot*) approach[17]. Sets of three local genotypes plus an improved variety were allocated randomly to farmers as incomplete blocks, maintaining spatial balance by assigning roughly equal frequencies of the genotypes. Each farmer also received an improved variety (*Asassa* in Tigray and Amhara, and *Hitosa* and *Ude* in Oromia), for a total of four plots per farmer. Trial size ranged from 0.4 m$^2$ to 1.6 m$^2$ depending on season and location. Field technicians provided guidance to farmers on the tricot approach prior the experiment. Farmers planted, managed and evaluated their own experiments. At the end of the growing season, farmers were visited by an enumerator and indicated the OA of genotypes by ranking the four entries that they received from best to worst, using pre-defined answer forms. Field technicians collected GY measures in farmers' plots after harvesting. Differently from the centralized trials, the OA was derived from the relative rankings of genotypes, as each farmer evaluated a different set of materials.

**Centralized trait data analysis**. All analyses were done in R[58]. GY$_{STATION}$ and OA$_{STATION}$ measured in centralized trials were used to derive best linear unbiased prediction (BLUP) values using the R package ASReml-R[59], treating locations as a fixed factor and all other factors as random. Full model details are reported in Supplementary Note 1. For the central comparison between benchmark and 3D-breeding, we used measures of GY$_{STATION}$ combined across seasons and locations (Eq. S1). Similarly, OA$_{STATION}$ in the central comparison represents OA values combined across genders and locations (Eq. S3). When relevant, GY$_{STATION}$ and OA$_{STATION}$ measures were split by location, season or gender (Supplementary Note 1). Broad sense heritability ($H^2$) and narrow-sense heritability ($h^2$) were derived for agronomic traits (Eq. S2) and farmers' OA (Eq. S4). Agreement between farmer gender groups in evaluating centralized station data was derived from a linear model fit. Spearman correlations between location specific BLUP values and farm performance were also computed.

**Decentralized trait data analysis**. For the analysis of the decentralized data, we used the Plackett–Luce model[60,61], using the R package PlackettLuce[54]. The implementation of Plackett–Luce model to analyze data from decentralized crop

variety trials is demonstrated by van Etten et al.[16]. Plackett–Luce is a rank-based model that follows the Luce's axiom of choice[61], which assumes that ranking order between every pair of options does not depend on the presence or absence of other options. The model estimates the *worth* parameter $\alpha$ which related to the probability ($P$) that one genotype $i$ wins against all other $n$ genotypes in set, and are obtained using the following equation:

$$P(i \succ \{j, ..., n\}) = \frac{a_i}{a_1 + ... + a_n} = \frac{a_i}{1} = a_i \quad (1)$$

**Implementation of the genomic prediction benchmark**. We established a benchmark that represents a centralized breeding approach enriched with farmer evaluations. We believe that this benchmark represents a realistic and competitive alternative to 3D-breeding. On-station involvement of farmers is not common practice but is increasingly conducted in association with breeding[14,18] and makes the benchmark more competitive. The stations selected for the benchmark were commonly used as breeding field trials for Amhara and Tigray regions of Ethiopia, and differ in altitude, temperature, rainfall, and soil[21]. Additional multilocation trials would typically occur in earlier stages of the breeding cycle. Centralized stations and farmer fields belong to the same agroecological zones of Ethiopia (Supplementary Fig. 11).

The benchmark was based on genomic prediction models and marker-based genetic relationship matrices computed on BLUP data with the package rrBLUP[62], a method widely used in breeding programs worldwide. To measure accuracy of genomic predictions, we calculated the Kendall's tau coefficient ($\tau$), a measure of similarity of rankings[63], between predicted values and observed values. The use of the $\tau$ metric, uncommon in breeding[64], allowed to compare accuracies with the 3D-breeding approach. A Pearson's correlation, the standard metric for genomic prediction accuracy, was also computed but did not show any relevant difference with the Kendall $\tau$. Also to provide a more coherent comparison with 3D-breeding, the benchmark was trained with ordinal rankings derived from absolute values of GY and OA measured in centralized trials, without showing any relevant difference from the training performed with absolute values.

The benchmark considered two main prediction scenarios. In the first scenario, prediction was restricted to the centralized experiment. In this scenario, the genomic prediction model was trained on $GY_{STATION}$ and $OA_{STATION}$ measured on the full set of 400 genotypes evaluated in 2012, and the training dataset was $GY_{STATION}$ measured in the same locations in 2013 on the subset of 41 genotypes that were also included in the 3D-breeding. In the second scenario, the benchmark was trained on combined $GY_{STATION}$ and $OA_{STATION}$ data in centralized trials and used to predict the test population of 41 genotypes measured in decentralized fields for $GY_{FARM}$ and $OA_{FARM}$. Mirroring the approach used in the 3D-breeding, the accuracy of genomic prediction in the second scenario was derived from a cross-validation approach averaging Kendall $\tau$ specific for Season 1, Season 2, and Season 3 using the square root of the sample size as weights[65].

The benchmark was tested with additional prediction scenarios considering different training and test populations, including: (i) without overlap between training and test samples, (ii) restricting the training to the subset of 41 genotypes selected for 3D-breeding, (iii) predicting $GY_{FARM}$ and $OA_{FARM}$ in decentralized fields stratified by their environmental distance from centralized stations.

**Implementation of the 3D-breeding**. The model representing the 3D-breeding approach was built with the data generated by the citizen science decentralized trials using Plackett–Luce Trees (PLT). This model includes covariates through recursive partitioning (successive binary splits based on covariate thresholds)[66]. We used PLT to analyze $OA_{FARM}$ and $GY_{FARM}$. DNA data from SNPs was added into the model as a prior using an additive matrix. Agroclimatic indices were used as covariates in the PLT model. Daily temperature and precipitation data were obtained from the NASA LaRC POWER Project (https://power.larc.nasa.gov/), using the R package nasapower[67]. The set of agroclimatic covariates was extracted for the vegetative, reproductive and grain filling phases and the whole growth period (from planting date to harvesting) in each observation point using the R package climatrends[68]. This resulted in 110 covariates.

To create a model that provides generalizable predictions across seasons with few covariates, we used blocked cross-validation (with seasons as blocks) combined with a forward selection[69]. We used the deviance values of each validation season to calculate an Akaike weight, which is the probability that a given covariate combination represents the best model[70]. We performed forward selection, using this combined Akaike weight as our selection criterion. The PLT models had a cut-off value of $\alpha = 0.01$ and a minimal partition size of 20 percent of the total dataset. The covariates selected under this procedure were the maximum night temperature (°C) during reproductive growth and the minimum night temperature (°C) during the vegetative growth. To compare the accuracy of the model representing 3D-breeding with the benchmark, we calculated the Kendall $\tau$ between observed rankings and predicted coefficients in farmer fields. To accommodate for the different number of observations derived from the benchmark and from decentralized fields, we run additional 3D-breeding scenarios trained with subsets of 75%, 50%, 25%, 15% and 5% of the decentralized plots to explore the prediction accuracy attainable by 3D-breeding with fewer observations. Details on the procedure are given in Supplementary Note 1.

**Generalization of the 3D-breeding**. To evaluate if the model obtained with the variable selection procedure retained predictive power across seasons, we simulated untested future seasonal climate with representative seasonal scenarios of past climate conditions by extracting the last 15 years of daily climate data derived from NASA POWER (2001–2015). We determined three windows for sowing dates in each growing season as the midpoints of equiprobable quantile intervals estimated from the observed planting dates in the data set. We predicted genotype performance for 15 seasons × 3 sowing dates (45 seasonal scenarios) for 1,200 random points generated across an alpha hull area within the range of the trials' coordinates. We averaged genotype probability of winning across these scenarios for each planting date interval, excluding the seasons used as testing data.

We calculated the reliability, the probability of outperforming a check variety[71]. We used the *worth* parameters from Plackett–Luce to determine the values of positive-valued parameters $\alpha_i$ associated with each genotype $i$, by comparing the *worth* from the check variety (*Asassa*, *Hitosa* and *Ude*, currently recommended for the mega-environment[24]) with the *worth* of the selected genotypes from 3D-breeding. These parameters ($\alpha_i$) are related to the probability ($P$) that genotype $i$ wins against all other $n$ genotypes in a set as shown in Eq. 1. To calculate the reliability of a genotype, we used Equation 2:

$$P(i \succ j) = \frac{a_i}{a_i + a_j} \quad (2)$$

**Environmental characterization of test sites and genotypes**. The agroecological zonation of Ethiopia was obtained by the Ethiopian Institute of Agricultural Research (EIAR)[72]. GPS coordinates of centralized stations and decentralized farmer fields were used to retrieve climatic data from NASA POWER. Temperature indices for covariates used in the PL model were retrieved for the growing seasons object of the study in the time span from sowing date and flowering dates as measured on-site. Climatic variables considered were the maximum night temperature (°C) during reproductive growth and the minimum night temperature (°C) during the vegetative growth, which showed to be the most relevant for the sampled data. A principal component analysis (PCA) was used to summarize and depict variation at test sites. Climatic distance of test sites was derived from a multidimensional scaling (MDS) of the multivariate climate dataset. For each of the two stations, climatic distance was computed with all farm sites. Wheat genotypes were split in cold adapted and warm adapted according to the altitude of their original sampling site with a one-tailed, unequal-variance t-test.

**Statistics and reproducibility**. Centralized experiments were run in two locations, for two seasons, on replicated plots for 400 genotypes for a total of 3,200 plots. The benchmark was run with different prediction scenarios considering separated and overlapping training and test populations and specified in the methods. Decentralized trials were performed on 1,165 farmer fields, with four plots per farmer field evaluated in ranking, for a total of 4,660 plots. Organizing the datasets relied on R packages data.table[73], caret[74], gosset[75], janitor[48], magrittr[76] and tidyverse[77]. Climatic variables were obtained using the packages climatrends[68] and nasapower[67]. Statistical analysis was performed using packages PlackettLuce[54], gosset[75] and qvcalc[78]. Spatial visualization was performed with the packages dismo[79], raster[80], sf[81] and smoothr[82]. Charts were produced using packages corrplot[83], ggplot2[84] and patchwork[85].

**Reporting summary**. Further information on research design is available in the Nature Research Reporting Summary linked to this article.

## Data availability
Data is available through Dataverse[56].

## Code availability
Code is available through Dataverse[56].

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

## Acknowledgements

We thank the farmers who evaluated the genotypes in both centralized and decentralized trials. We also thank Olga Spellman (Alliance of Bioversity International and CIAT) for English editing, and Rebecca Nelson, Geoffrey Morris, Roberto Buizza and Martina Occelli for the critical discussions and valuable insights. We thank the three anonymous reviewers for their thoughtful revisions. This work was implemented as part of the CGIAR Research Program on Climate Change, Agriculture and Food Security (CCAFS), which is carried out with support from the CGIAR Trust Fund and through bilateral funding agreements. For details, please visit https://ccafs.cgiar.org/donors. The work of K.d.S. and S.Ø.S. was supported by The Nordic Joint Committee for Agricultural and Food Research (grant num. 202100-2817). The work of M.D.A., M.E.P., Y.G.K. and B.F.L. was supported by the Doctoral School for Agrobiodiversity at Scuola Superiore Sant'Anna.

## Author contributions

M.D.A., J.v.E., J.P., K.d.S., S.Ø.S., J.L.J., C.F. and M.E.P. designed research; C.F., Y.G.K., D.K.M., B.F.M., performed field research; M.D.A., J.P. and K.d.S. analyzed data; K.d.S., M.D.A., J.v.E. and J.L.J. wrote the paper; S.Ø.S. and J.P. commented on all versions of the manuscript and contributed by suggesting novel additional analyses and interpretations.

## Competing interests

The authors declare no competing interests.
