## [Peer Review File · Communications Biology]

Reviewers' Comments:

Reviewer #1:

Remarks to the Author:

Manuscript#: COMMSBIO-21-0076-T

The manuscript "Data-driven decentralised breeding increases genetic gain in a challenging crop production environment" seems sufficiently promising to be published in *Communication Biology*. However, the authors need to provide significant revisions to the original manuscript in order to improve the technical details and the grammatical misspellings.

Major Comments

Lines 290 to 292:

1) How were the locations of the decentralized plots selected? Do they represent all the Ethiopian regions where wheat crop is usually grown?

2) Have the farmers been trained to evaluate the genotypes using the Likert' scales for overall appreciation? This is really important if different genotypes sets were evaluated by distinct farmers.

Lines 342 to 346:

3) How the environmental differences were modeled to predict the genotypes performances across distinct locations and years? It is not clear how the environmental distances were included in the genomic prediction models.

Figures:

4) Figure 1: What are the meanings of Var A, Var B, Var C and Var D? These terms are not mentioned in other parts of the manuscript.

5) Figure 2A: The legend should be positioned out of the plotting area, otherwise it is confusing.

Supplementary Materials:

6) It is not clear why the effect of years was considered as random and the effect of locations as fixed. Please, explain these assumptions.

7) The equations in the supplementary material should be enumerated such as s_1 , $s_{1.1}$, and so on. Moreover, these equations were not cited in the manuscript and it is not clear the cases where they were used.

Minor Comments

Line 307: "ASReml" should be "ASReml-R"

Line 290: "decentralised" should be decentralized.

Reviewer #2:

Remarks to the Author:

Decentralized plant breeding may be key to ensuring that marginalized small farmers have access to plant varieties that are adapted to their environments and diverse needs. This paper provides an interesting case for enabling such a system through data driven decentralized breeding in which the authors compared centralized and decentralized managed plots to determine farmer appreciation of varieties and harness their knowledge about adaptation in different environments (GxExM) to inform genomics assisted selection. The paper is novel in this sense and contributes a promising tool for the challenging arena of limited farmer adoption of new varieties in marginal environments. The authors were able to crowdsource variety information from farmers to increase the accuracy of a benchmark representing genomic selection. They propose the tool as a means to leverage the diversity in farmers' fields to complement conventional breeding. This is meaningful research that could benefit smallholder farmers, though care should be taken to make sure that diversity is not lost and farmers' knowledge is duly acknowledged and compensated. I will limit my

comments to the value, organization, and descriptive aspects of the methods, as my speciality is not genetics, rather crop ecology and participatory plant breeding. I cannot comment on the appropriateness or validity of the statistical analysis.

There are several areas that need to be addressed to make this paper more cohesive, accessible, and justified in its interdisciplinary approach. While the journal is clearly focused heavily on genetics and the paper is written with that orientation, the paper is actually bridging development science with farmer knowledge and science. As it is, it is presented exclusively for a crop geneticist audience and this does a disservice to the paper, its potential impact, and the potential audience it could reach. The authors should consider who their intended audience is and their desired goal for the paper and adjust accordingly. My suggestions are focused on making it a bit more accessible, either way the authors go.

Organization

The paper has only a section labeled materials and methods with some of the methods summarized earlier in the paper (71-75~) and some missing information in the methods section itself. The rest of the sections have topic subheadings but overall it does not flow in a way that the reader can grasp the sequence of activities and the importance of them. An introduction and results section would help, but the reviewer is open to different forms of organization as long as they are clear. I suggest better section headings, a more logical framework for presenting the paper (maybe even a diagram) background information on a few topics like genomic benchmarks (a sentence or two), why gender matters (you analyze it but don't talk about it), and crowdsourcing. The authors have limited space but orienting the readers a bit more cohesively will make for a better paper.

Due to a lack of headings, I felt like I was reading methods within results or discussion and found it hard to know when I was reading some of your results vs. info from other work. Please organize this paper in a way that is accessible, and the reader doesn't have to jump between pages to really understand what is going on.

Other comments:

Line 82: This is confusing when the idea is to evaluate whether 3D-breeding could complement genomic breeding by increasing accuracy. I thought they were supposed to complement each other by using 3D breeding to enhance genomic assisted breeding.

Methods

There is discussion of crowd-sourcing and citizen science which leads back to some other papers published by the authors. However, in the methods it is not mentioned how you crowd-sourced this data. Is it through phones? Is it through surveys/technicians visiting farmers' fields to collect their preference data? How was the data collected? Why was there such variation in the number of farmers each year?

From the earlier section (line 71-75~) it's clear that farmers evaluated 400 fully genotyped durum wheat lines on-station. (It would be nice if it said this in the methods section). This was narrowed down to 41 for the decentralized. In my experience with PPB and PVS, this number of lines for farmer evaluation is very large and perhaps unprecedented. 30 farmers evaluated all 400? At the same time? How was the OA established? Were there any guidelines that were co-developed or was it just as the farmers' overall appreciation, however they viewed that? Was it only agronomic traits assessed? Are these farmers the station/researchers worked with regularly? How were they chosen?

How were the farmers chosen for the station work? And the decentralized fields?

You do some analysis that differentiates men and women. But you don't discuss it or give a clear indication of what that means, or even better yet, why did you look at both the OA of men and women? And did you also do women's fields for the decentralized trials? What does it mean that there were no differences between men and women in overall appreciation? Were they only measuring field factors? Was there any observations or scoring related to processing and end-use?

Research has shown that this is where much of the differences come in, or in different fertility levels in fields. Please include background info on why you differentiated gender, what the findings were, and what they mean.

If your purpose is to reach the marginalized farmers, how did you start that process in this work? How did you make sure those engaged were the marginalized? I realize this is testing a concept, but you engaged with 1165 farmers. How you engaged with them and identified them is relevant. If this information is cited in another work, then please indicate so.

In "The way forward" you mention an "open question" of how to motivate farmers to participate in the evaluation of material. How did you motivate them in this study? It seems an important part of the methods and helpful for future work in this area.

In this section, I appreciate another one of the "open questions" that briefly mentions how to share benefits deriving from the utilization of farmers' knowledge to produce new varieties. This is a very important point that cannot be emphasized enough, and the diversity that farmers have been able to maintain and enable access to for breeders is an important angle as well that needs to be at the center of such endeavors.

The premise of this paper, to some degree, is that farmers have intrinsic knowledge of varieties well suited to their environment (which has been elusive for plant breeders) and that by harnessing this knowledge, plant breeding can better serve marginalized smallholders. In the way forward section you indicate that this method could also be used for high input systems. This could be an interesting application, but adoption of new varieties in such environments is not limited, and the high input options make broadly adapted varieties fairly well suited to these environments. Are you suggesting that it will make them even better suited? Or maybe the point is to identify varieties for such systems that don't require as much inputs in order to be high yielding? That might be an interesting angle on sustainability and application of 3d-breeding.

Discussion/Conclusions

What ramifications does 3D-breeding have for diversity? If farmers have so many varieties (that according to current rhetoric are destined to be obsolete with climate change) then what happens if breeding can develop better varieties based on the combined knowledge of farmers and breeders? What are some considerations for conservation or maintenance in place for the diversity that does exist in wheat grown in Ethiopia, or any other system this could be transferred to? Such a discussion is beyond the scope of your paper but may be part of "the way forward" important questions section.

Reviewer #3:

Remarks to the Author:

The paper describes a new innovative breeding approach named 3D-breeding in which the authors combine genomics, environmental and farmer's knowledge factors in a single analytical frame that exploits phenotypic records in farmer's fields (decentralized trials) instead of research center trials (centralized trials). Despite the simplicity of the idea, I think 3D-breeding will open a new gate in the field of genomic prediction and will have direct industrial impact on reducing efforts and costs while increasing the size of reference data, environmental / management variation, and the overall efficiency of breeding programs. Such approach is timely needed, and I would love to see the paper published. While it is very convincing for me in theory that largescale 3D-breeding should provide better prediction compared to the centralized breeding due to the exploitation of diverse management and environmental data, the design of the present study does not provide fair comparison with centralized breeding. Moreover, some of the results presented in the manuscript seem to be contradictory with what have been shared on Dataverse (unless if I misunderstood something). The below comments can be useful to improve the content of the manuscript:

Major comments:

1- Lines 292-296: On Dataverse (file: 4_DataFile_DecentralizedBreeding_DurumWheat.xlsx), there are 71 genotypes plus the 3 cultivars, please clarify this point. There were 41 genotype names in Figure S14 including the three recommended genotypes for the mega-environment (Asassa:

341ET_D; Hitosa: 390ET_D; Ude: 344ET_D). This seems to be different from what described in the paragraph in which I understood that there were 41 genotypes plus 3 cultivars. Please reword.

2- For each prediction scenario, the reference and the validation populations and their sizes were not always fully described. For example, in lines 342 to 349, it is not clear if the authors used the subset of the 41 genotypes in the reference or not. I would recommend adding a supplementary table describing different scenarios with the sizes of their training and validation populations.

3- The reference population for the centralized trials consists of 400 genotypes scored in two seasons in two stations (expecting a total of 1,600 datapoints) while the top 41 genotypes of them were selected for the decentralized trials with a total of 3,252 datapoints (inferred from the file: 4_DataFile_DecentralizedBreeding_DurumWheat.xlsx; for these 41 genotypes). Thus, there is no wonder why 3D-Breeding performed better than the centralized breeding in your experiment given that:

a) Your decentralized reference population is exactly the same as the validation population (41 genotypes) compared to the centralized populations that involved 400 different genotypes

b) The size of the decentralized reference population seems to be larger than the size of the centralized reference population

I acknowledge that it will be impossible to convince the farmers to grow the remaining 359 genotypes with lower performance in the centralized trials, but another design should have been carried out to better synchronize both comparisons. At least similar reference sizes should have been used if not already done. Another possible way is to use a subset of the station genotypes (other than selected 41 genotypes) as a validation population to compare predictability of the station and the farm references.

4- Looking at Figure S14, the best performing genotypes for the three nodes are 341ET_D, 344ET_D, 390ET_D, 441ET_D and 033ET_D. The first three of these are the three recommended genotypes for the mega-environment (Asassa, Ude and Hitosa; as described in the file:

4_DataFile_DecentralizedBreeding_DurumWheat.xlsx). This makes the results of Figure 2A very odd. Moreover, in the same file, Asassa and Hitosa showed the lowest average "farmer_rank" with 1.74 and 1.67; respectively, which is against what is described in lines 155 to 157.

5- I would love to see a prediction scenario in which both centralized and decentralized data were used together in the training population, this may further have higher prediction accuracy

Minor comments:

Line 24: I guess you have 1,165 farms in each one you have four plots. Please reword throughout the manuscript to avoid confusion

I would prefer to calculate heritability and genetic correlation with GREML method, e.g. using GCTA or MTG2 software.

Table S2: The heritability of OA is very high. I assume that the farmer's selection in Ethiopia will be highly affected by heading date. Appreciate if you can check the correlation between OA and other recorded traits, especially phenology related traits.

Is there any previous study on the level of the within genetic diversity for the studied landraces? And how many seeds you used for 90K genotyping per landrace?

If the authors have information about the age of the farmers, I think it worth to do a correlation based on the age other than the gender as this will reflect the experience of the farmer.

Here below, we include a point-by-point response to reviewers. Our text is in red type

Referee expertise:

Referee #1: genomic prediction in crops

Referee #2: smallholder farms, crop ecology

Referee #3: crop breeding, genomics

Reviewers' comments:

Reviewer #1 (Remarks to the Author):

Manuscript#: COMMSBIO-21-0076-T

The manuscript “Data-driven decentralised breeding increases genetic gain in a challenging crop production environment” seems sufficiently promising to be published in Communication Biology. However, the authors need to provide significant revisions to the original manuscript in order to improve the technical details and the grammatical misspellings.

Major Comments

Lines 290 to 292:

1) How were the locations of the decentralized plots selected? Do they represent all the Ethiopian regions where wheat crop is usually grown?

Centralized and decentralized locations were representative for the dominant wheat growing zones in Ethiopia, Amhara, Tigray and Oromia (see e.g. <https://doi.org/10.1080/23311932.2020.1778893>) and belong to the same agroecological zones (Fig. S3). The selection of decentralized locations was conducted as follows: we started by identifying the districts on which the 3D-breeding experiment was to be conducted based on prior experience of wheat production. Then, together with district agriculture offices we identified the villages most representative of wheat production. Finally, we identified the farmers together with local development agents. In the selection of the farmers, we both

included those regarded as virtuous model and those that were not. We clarify this point in the revised Materials and Methods section (from L135):

“Farms were selected in areas representative for wheat growing in Ethiopia, based on previous history of cultivation of the crop (Fig. S2). Individual farmers were engaged via local agricultural offices and selected based on their willingness to participate and on the following criteria: i) being wheat growers, ii) owning their land, iii) living in the village all year. No financial incentive was given to farmers besides the opportunity to test new varieties and harvest from the decentralized varietal plots. Selected farms were representative of the agroecological zones of the centralized fields (Fig. S3).”

2) Have the farmers been trained to evaluate the genotypes using the Likert’ scales for overall appreciation? This is really important if different genotypes sets were evaluated by distinct farmers.

In centralized plots:

Farmers were trained in participatory varietal evaluation prior the experiment using focus group discussions. During the Likert scoring for overall appreciation, they were accompanied by a researcher who guided the evaluation and collected OA values for individual farmers. To clarify the evaluation procedure, we added a supplementary video in which we can see a group of women farmers performing the evaluation. Full details are given in Mancini et al 2017. This is now clarified in materials and methods (from L120):

“The participants provided appraisal with Likert scales²³ to genotypes for overall appreciation ($OA_{STATION}$)^{20,21}, with 1 being worse and 5 the best. Prior the experiment, farmers were involved in focus group discussions and trained on how to perform the evaluation²¹. During the evaluation, farmers were divided in gender-homogenous groups of 5 people, were introduced in the field from random entry points, and were accompanied plot by plot by a researcher who guided the evaluation and collected $OA_{STATION}$ values from individual farmers (Supplemental Movie 1).”

The Supplemental Movie is fully described in the Supplementary Note:

“Movie S1: Participatory evaluation exercise in centralized fields. A group of five women farmers is engaged in evaluating a wheat plot by an enumerator (his back to the camera). The women farmers provide Likert scores counting on their fingers, and verbally describe the reasoning behind their score to the enumerators. The evaluation featured in the movie was not part of the data collection featured in this paper, but it is meant for display purposes. In the real experiment, farmers provided scores without other members of the farmer group seeing it. Enumerators did not collect verbal description of choice processes. All details are given in Mancini et al. (2017)¹.”

In decentralized plots:

Farmers were trained by field technicians prior the evaluation and during focus group discussions. Concurrently, they were trained about the plot size to use, that was standardized upfront. However, one must point out that that tricot evaluation of varieties is quite easy to perform, as it just involves a ranking of varieties. The outcome of the decentralized evaluation was collected by dedicated enumerators, so that one enumerator was appointed a village (15 famers on average) and collected farmers’ rankings and agronomic traits on plot. This is now clarified in the materials and methods (from L149):

“Field technicians provided guidance to farmers on the tricot approach prior the experiment. Farmers planted, managed and evaluated their own experiments. At the end of the growing season, farmers were visited by an enumerator and indicated the OA of genotypes by ranking the four entries that they received from best to worst, using pre-defined answer forms.”

Lines 342 to 346:

3) How the environmental differences were modeled to predict the genotypes performances across distinct locations and years? It is not clear how the environmental distances were included in the genomic prediction models.

Environmental distances were used to interpret the results, detecting possible shortcomings in the centralized training to target specific farm locations. The genomic prediction is trained on BLUPs computed using location and year as fixed and random effects as the benchmark

would represent a prediction model commonly used in crop breeding approaches: the G-BLUP (equivalent to RR-BLUP). We clarified this point in text (L191):

“The benchmark was based on genomic prediction models and marker-based genetic relationship matrices computed on BLUP data with the package rrBLUP²⁹, a method commonly used in breeding programs worldwide.”

And (L212)

“The benchmark was tested with additional prediction scenarios considering different training and test populations, including: i) without overlap between training and test samples, ii) restricting the training to the subset of 41 genotypes selected for 3D-breeding, iii) predicting GY_{FARM} and OA_{FARM} in decentralized fields stratified by their environmental distance from centralized stations.”

Figures:

4) Figure 1: What are the meanings of Var A, Var B, Var C and Var D? These terms are not mentioned in other parts of the manuscript.

Apologies for the confusion. These are hypothetical varieties used to exemplify how the different approaches will recommend varieties based on evidence reported in VanEtten et al 2019 and in our manuscript. We expect that the decentralized approach will recommend a larger set of varieties as it is able to capture the GxE interaction at a finer scale. This is now clarified in the figure caption (L710):

“Fig. 1. Centralized breeding (**A**) derives recommendations from breeders’ evaluation and possibly participatory assessments in a limited set of stations, using genomics to accelerate the production of varieties that are eventually recommended with low spatial resolution. The plot to the right shows the coarse recommendation space of two hypothetical varieties, Var A and Var B. This system may become more efficient if complemented by 3D-breeding (**B**), a decentralized approach where the best candidate genotypes are tested by farmers in small, blinded and randomized sets. 3D-breeding produces scalable solutions that can be linked to genomics, farmers’ knowledge and environmental data, to enhance the local adaptation of the

resulting varieties and tailor their recommendation to the landscape. This is represented in the plot to the right by the refined recommendation space of hypothetical varieties Var A, Var B, Var C and Var D.“

5) Figure 2A: The legend should be positioned out of the plotting area, otherwise it is confusing.

We would prefer not to move the legend outside the plot in panel A, otherwise it may suggest that it also applies to panels B and C. To address the reviewer’s point and reduce confusion, we added a shading to the legend in panel A. See figure below:

Supplementary Materials:

6) It is not clear why the effect of years was considered as random and the effect of locations as fixed. Please, explain these assumptions.

We considered year effects as random as they could not be controlled experimentally, while location effects are fixed because they are included by design. This is now clarified in the supplementary note:

“The model considers locations a fixed factor included by experimental design. The year effect is considered random because it cannot be controlled experimentally.”

7) The equations in the supplementary material should be enumerated such as s1, s1.1, and so on. Moreover, these equations were not cited in the manuscript and it is not clear the cases where they were used.

We clarified the naming of supplemental equations (Eq. s1 to Eq. s4) and cited their usage in the Materials and Methods section (L160):

“For the central comparison between benchmark and 3D-breeding, we used measures of $GY_{STATION}$ combined across seasons and locations (Eq. S1). Similarly, $OA_{STATION}$ in the central comparison represents OA values combined across genders and locations (Eq. S3). When relevant, $GY_{STATION}$ and $OA_{STATION}$ measures were split by location, season or gender (Supplementary Note 1). Broad sense heritability (H^2) and narrow-sense heritability (h^2) were derived for agronomic traits (Eq. S2) and for farmers’ appreciation (Eq. S4).”

Minor Comments

Line 307: “ASReml” should be “ASReml-R”

Fixed

Line 290: “decentralised” should be decentralized.

Fixed

Reviewer #2 (Remarks to the Author):

Decentralized plant breeding may be key to ensuring that marginalized small farmers have access to plant varieties that are adapted to their environments and diverse needs. This paper provides an interesting case for enabling such a system through data driven decentralized breeding in which the authors compared centralized and decentralized managed plots to determine farmer appreciation of varieties and harness their knowledge about adaptation in different environments (GxExM) to inform genomics assisted selection. The paper is novel in this sense and contributes a promising tool for the challenging arena of limited farmer adoption of new varieties in marginal environments. The authors were able to crowdsource variety information from farmers to increase the accuracy of a benchmark representing genomic selection. They propose the tool as a means to leverage the diversity in farmers' fields to complement conventional breeding. This is meaningful research that could benefit smallholder farmers, though care should be taken to make sure that diversity is not lost and farmers' knowledge is duly acknowledged and compensated. I will limit my comments to the value, organization, and descriptive aspects of the methods, as my speciality is not genetics, rather crop ecology and participatory plant breeding. I cannot comment on the appropriateness or validity of the statistical analysis.

There are several areas that need to be addressed to make this paper more cohesive, accessible, and justified in its interdisciplinary approach. While the journal is clearly focused heavily on genetics and the paper is written with that orientation, the paper is actually bridging development science with farmer knowledge and science. As it is, it is presented exclusively for a crop geneticist audience and this does a disservice to the paper, its potential impact, and the potential audience it could reach. The authors should consider who their intended audience is and their desired goal for the paper and adjust accordingly. My suggestions are focused on making it a bit more accessible, either way the authors go.

Organization

The paper has only a section labeled materials and methods with some of the methods summarized earlier in the paper (71-75~) and some missing information in the methods section itself. The rest of the sections have topic subheadings but overall it does not flow in a way that the reader can grasp the sequence of activities and the importance of them. An introduction and results section would help, but the reviewer is open to different forms of

organization as long as they are clear. I suggest better section headings, a more logical framework for presenting the paper (maybe even a diagram) background information on a few topics like genomic benchmarks (a sentence or two), why gender matters (you analyze it but don't talk about it), and crowdsourcing. The authors have limited space but orienting the readers a bit more cohesively will make for a better paper. Due to a lack of headings, I felt like I was reading methods within results or discussion and found it hard to know when I was reading some of your results vs. info from other work. Please organize this paper in a way that is accessible, and the reader doesn't have to jump between pages to really understand what is going on.

We critically rearranged the manuscript sections to improve accessibility of the relevant information. We rearranged the materials and methods section and combined the results and discussion section as per the journal guidelines. We added headings to the guide the reader throughout the main sections of the paper, focusing results and discussion to specific bits of information supporting our main findings. Headings of the Results and Discussion section include:

- Performance of centralized breeding based on genomic prediction and farmers' traditional knowledge
- Benchmark: using centralized measures to predict performance in farmer fields
- 3D-breeding provides higher prediction accuracy than the benchmark
- Superior genotype selection with 3D-breeding is consistent across seasons
- Implications for rethinking breeding programs
- Potential of 3D-breeding for challenging cropping environments

We also included a paragraph to the last part of the Introduction section, rephrasing it to clearly explain the framework of the paper (L73).

“Here, we collected data from the genotyping and phenotyping of 400 wheat varieties in centralized stations commonly used for varietal selection in Ethiopian highlands. We then selected and distributed a subset of 41 genotypes as packaged sets containing incomplete blocks of three genotypes, plus one commercial variety to each of 1,165 farmers located in the same breeding mega-environment. We tested 3D-breeding against a competitive benchmark that represents breeding based on a genomic prediction model trained on centralized stations to predict varietal performance in farmers’ decentralized fields. We focused on grain yield (GY) and farmers’ overall appreciation (OA) of wheat genotypes, which were both recorded in centralized and decentralized trials. To establish the benchmark, we used a genomic prediction model trained on data measured in stations to predict wheat GY and OA in farmer fields (Fig. 1A). We then developed 3D-breeding to move the selection to farmer fields, predicting GY and OA in farmer fields using a decentralized approach (Fig. 1B). Comparing side by side the accuracy of the two methods, we found that that 3D-breeding could increase prediction accuracy in challenging environments.”

Other changes are too extensive to be reported here but are highlighted with the track change feature throughout the manuscript.

Other comments:

Line 82: This is confusing when the idea is to evaluate whether 3D-breeding could complement genomic breeding by increasing accuracy. I thought they were supposed to complement each other by using 3D breeding to enhance genomic assisted breeding.

We clarified this point here and elsewhere in text. We see 3D-breeding as complementary to centralized breeding, but in order to prove its value we had to compare it against a competitive benchmark. The new wording is not “3D-breeding competing against centralized breeding” but rather “3D-breeding compared with a competitive benchmark representing centralized breeding”

Methods

There is discussion of crowd-sourcing and citizen science which leads back to some other

papers published by the authors. However, in the methods it is not mentioned how you crowd-sourced this data. Is it through phones? Is it through surveys/technicians visiting farmers' fields to collect their preference data? How was the data collected? Why was there such variation in the number of farmers each year?

We added all the relevant information in the revised Materials and Methods section (see answer to Reviewer#1). Data was collected by technicians visiting the farmer fields at the end of the season and recording data on preformatted forms (L150):

“Farmers planted, managed and evaluated their own experiments. At the end of the growing season, farmers were visited by an enumerator and indicated the OA of genotypes by ranking the four entries that they received from best to worst, using pre-defined answer forms. Field technicians collected grain yield (GY_{FARM}) in farmers' plots after harvesting.”

The reason why there was such variation in number of farmers year by year is that not all the participating farmers could be reached in one season. Rather, the experiment expanded throughout the years including different communities. This is now clarified in the Materials and Methods section (L140):

“Differences in number of fields by season are due to availability of farmer communities to join the study.”

Additionally, some of the data could not be used in this manuscript as relied on genotypes for which no genomic information was available (see response to Reviewer#3)

From the earlier section (line 71-75~) it's clear that farmers evaluated 400 fully genotyped durum wheat lines on-station. (It would be nice if it said this in the methods section). This was narrowed down to 41 for the decentralized. In my experience with PPB and PVS, this number of lines for farmer evaluation is very large and perhaps unprecedented.

Indeed, this number of genotypes is high as compared with previous PPB and PVS literature. In the centralized fields, the evaluation took about a full week of work in each location. This

reflects the unique quantitative approach that we use in PVS. Full reports of the size of the experiments and data exploration are reported in Mancini et al 2017 and Kidane et al 2017.

In the decentralized fields, the high number of evaluations is achieved by parceling the work across hundreds of farmers, each performing a simple evaluation of an incomplete block of three randomized lines plus one check variety (current recommendation for the area).

The methods section now includes a “Statistics and reproducibility” section detailing the experiment size (L276):

“Centralized experiments were run in two locations, for two seasons, on replicated plots for 400 genotypes for a total of 3,200 plots. The benchmark was run with different prediction scenarios considering separated and overlapping training and test populations as specified in the methods. Decentralized trials were performed on 1,165 farmer fields, with four plots per farmer field evaluated in ranking, for a total of 4,660 plots.”

30 farmers evaluated all 400? At the same time? How was the OA established? Were there any guidelines that were co-developed or was it just as the farmers’ overall appreciation, however they viewed that? Was it only agronomic traits assessed?

Yes; 30 farmers in each location evaluated all 400 genotypes in replicate, for a total of 800 plots. The evaluation occurred simultaneously in groups of five farmers conducted in the field from random entry points. The OA was established in focus group discussions as the farmers’ overall appreciation, however they viewed that. Farmers evaluated the field at flowering time so that their OA of any genotype is linked to its appearance at that growth stage. We also added a video showing an exercise of evaluation as Supplemental Movie 1 (see response to Reviewer #1). We provided all these answers in the revised paragraph in the Methods section (L116):

“In each location, 15 men and 15 women who were experienced smallholder farmers growing durum wheat were invited to evaluate plots during the 2012 season. The evaluation was conducted at flowering time and independently in each experimental station, for a total of 60 farmers involved. The farmers had no previous knowledge of the genotypes included in the study to prevent bias in the evaluations. The participants provided appraisal with Likert

scales²³ to genotypes for overall appreciation ($OA_{STATION}$)^{20,21}, with 1 being worse and 5 the best. Prior the experiment, farmers were involved in focus group discussions and trained on how to perform the evaluation²¹. During the evaluation, farmers were divided in gender-homogenous groups of 5 people, were introduced in the field from random entry points, and were accompanied plot by plot by a researcher who guided the evaluation and collected $OA_{STATION}$ values from individual farmers (Supplemental Movie 1).”

Are these farmers the station/researchers worked with regularly? How were they chosen? How were the farmers chosen for the station work? And the decentralized fields?

Farmers evaluating stations were chosen due to their wheat growing experience (L116). They belong to communities with whom local research stations work regularly. Farmers in the decentralized fields were chosen following different criteria (see responses to Reviewer#1), now clarified in the Materials and Methods (L133):

“Farms were selected in areas representative for wheat growing in Ethiopia, based on previous history of cultivation of the crop (Fig. S2). Individual farmers were engaged via local agricultural offices and selected based on their willingness to participate and on the following criteria: i) being wheat growers, ii) owning their land, iii) living in the village all year. No financial incentive was given to farmers besides the opportunity to test new varieties and harvest from the decentralized varietal plots.”

You do some analysis that differentiates men and women. But you don't discuss it or give a clear indication of what that means, or even better yet, why did you look at both the OA of men and women? And did you also do women's fields for the decentralized trials? What does it mean that there were no differences between men and women in overall appreciation? Were they only measuring field factors? Was there any observations or scoring related to processing and end-use? Research has shown that this is where much of the differences come in, or in different fertility levels in fields. Please include background info on why you differentiated gender, what the findings were, and what they mean.

From previous research, we expected to see a gender effect on OA in centralized stations. This is common knowledge in PPB and PVS, as men and women are engaged in different

parts of the cropping endeavor; in the case of Ethiopia, women are more concerned with marketing of seeds and men are more focused on field traits, so we might expect to see some difference in their evaluations (see e.g. Mancini et al 2017). Due to the extensive sampling for the tricot experiment we could not target a balanced number of men and women. Moreover, sex disaggregation used in isolation may overlook other issues that may correlate but also intersect with gender, such as income, occupation, marital status, ethnicity, age, or social status, and thus result in bias in the data (see e.g. <https://doi.org/10.4160/23096586RTBWP20212>). While we acknowledge the importance of traits related to processes, taste or end use, we could not measure those traits in the PVS scheme that we followed (see previous response about evaluations in centralized locations). Indeed, farmers could only score what they could see in the field at flowering time. To acknowledge the reviewer point, we added a more detailed explanation of gender aspects of PVS (L301):

“Previous studies showed that men and women may prioritize different traits depending on their role in the farming activity, from cropping to marketing of products^{55,56}. In our study, gender differences in $OA_{STATION}$ scoring are reflected by different H^2 achieved by men (0.84) and women (0.67), with a more marked difference in Hageselam (Table S2). Still, men and women provided consistent evaluations (Fig. S5). This is in line with tricot observations reporting that gender have low overall effect on varietal choice¹⁷ and shows that farmer scores are reliable measures of genotypes performance. Indeed, we found that $OA_{STATION}$ was a better predictor than $GY_{STATION}$ to capture both $OA_{STATION}$ and $GY_{STATION}$, in the following season, including when disaggregated by gender (Fig. S6). Previous studies explored the relation between OA and agronomic performance of wheat, showing that farmers’ appreciation was positively correlated to yield, seed size, biomass, and negatively correlated with time to flowering and time to maturity^{20,21}.”

We agree that future research should further investigate gender in crowdsourcing, and we added this information in the Conclusion section (L486):

“The expansion of the design with the addition of further testing seasons and local management conditions may allow to highlight drivers of local performance of genotypes beyond temperature⁸⁵. Further 3D-breeding studies may opt to stratify participants for

socioeconomic features of interest, including gender, age, or income, to fully characterize traditional knowledge in its many dimensions. Other traits may be considered to leverage farmers' traditional knowledge, such as those related to processes, taste or end use. Ideally, 3D-breeding could be combined with conventional, centralized breeding to improve the training of prediction models to address local adaptation.”

If your purpose is to reach the marginalized farmers, how did you start that process in this work? How did you make sure those engaged were the marginalized? I realize this is testing a concept, but you engaged with 1165 farmers. How you engaged with them and identified them is relevant. If this information is cited in another work, then please indicate so.

This comment stems from a poor choice of wording from our side, now corrected. The adjective “marginal” was intended to describe low-input agriculture, *i.e.* with characterized by low yields and low resilience. We never intended to refer to farmers as “marginalized”. Farmers sampled in the study are a random sample of smallholder farmers representative of wheat growing in the area. We removed the adjective “marginal” throughout the paper and we use instead the definition “challenging”

In “The way forward” you mention an “open question” of how to motivate farmers to participate in the evaluation of material. How did you motivate them in this study? It seems an important part of the methods and helpful for future work in this area.

In this section, I appreciate another one of the “open questions” that briefly mentions how to share benefits deriving from the utilization of farmers' knowledge to produce new varieties. This is a very important point that cannot be emphasized enough, and the diversity that farmers have been able to maintain and enable access to for breeders is an important angle as well that needs to be at the center of such endeavors.

Farmers did not receive any compensation besides the possibility to keep the harvest from the tricot plot. This is now clarified in the Materials and Methods section. Farmers are

interested in testing genetic materials because they can identify new preferable variables. They can keep the seeds afterwards, which may provide superior to what they had access to. Other studies, like the Beza et al 2017 now included in citations, explored motivation in participatory research (see Fig. 2 in that paper). We expand in the reviewer remarks in a new paragraph in the section about 3D-breeding potential at the end of the Results and Discussion section (L440):

“Both in centralized stations and in decentralized fields, we found that farmers were eager to participate without material compensation. Farmers seek access to new genetic materials that they could not access otherwise, in exchange for the minimum investment of running small plots and providing a synthetic evaluation at the end of the season in the case of the *tricot* evaluation¹⁷. This happens even if some of the provided varieties may not be adapted to their growing environment. Previous studies showed that farmers perceive as beneficial the interaction with experts and the sharing of information⁷⁴. Benefit to farmers may exceed the immediate access to improved technology, if the deeds to reconcile farmers’ and breeders’ rights in plant variety protection succeed⁷⁵.”

The premise of this paper, to some degree, is that farmers have intrinsic knowledge of varieties well suited to their environment (which has been elusive for plant breeders) and that by harnessing this knowledge, plant breeding can better serve marginalized smallholders. In the way forward section you indicate that this method could also be used for high input systems. This could be an interesting application, but adoption of new varieties in such environments is not limited, and the high input options make broadly adapted varieties fairly well suited to these environments. Are you suggesting that it will make them even better suited? Or maybe the point is to identify varieties for such systems that don’t require as much inputs in order to be high yielding? That might be an interesting angle on sustainability and application of 3d-breeding.

We meant exactly that 3D-breeding could identify varieties for such systems that don’t require as much inputs in order to be high yielding. To clarify this point, we added this sentence (L433):

“In these settings, 3D-breeding could contribute to the identification and development of varieties with higher local adaptation, reducing the need of external inputs to achieve desired yields.”

Discussion/Conclusions

What ramifications does 3D-breeding have for diversity? If farmers have so many varieties (that according to current rhetoric are destined to be obsolete with climate change) then what happens if breeding can develop better varieties based on the combined knowledge of farmers and breeders? What are some considerations for conservation or maintenance in place for the diversity that does exist in wheat grown in Ethiopia, or any other system this could be transferred to? Such a discussion is beyond the scope of your paper but may be part of “the way forward” important questions section.

We agree and mentioned this discussion in the conclusion section (L492):

“Once new varieties are developed through the crowdsourced combination of breeders’ and farmers’ knowledge, future research shall focus on the potential impact of these methods on conservation and use of traditional agrobiodiversity both *in situ* and beyond the local environments in which it was developed.”

Reviewer #3 (Remarks to the Author):

The paper describes a new innovative breeding approach named 3D-breeding in which the authors combine genomics, environmental and farmer’s knowledge factors in a single analytical frame that exploits phenotypic records in farmer’s fields (decentralized trials) instead of research center trials (centralized trials). Despite the simplicity of the idea, I think 3D-breeding will open a new gate in the field of genomic prediction and will have direct industrial impact on reducing efforts and costs while increasing the size of reference data, environmental / management variation, and the overall efficiency of breeding programs. Such approach is timely needed, and I would love to see the paper published. While it is very convincing for me in theory that largescale 3D-breeding should provide better prediction

compared to the centralized breeding due to the exploitation of diverse management and environmental data, the design of the present study does not provide fair comparison with centralized breeding. Moreover, some of the results presented in the manuscript seem to be contradictory with what have been shared on Dataverse (unless if I misunderstood something). The below comments can be useful to improve the content of the manuscript:

Thank you for considering our approach of interest and for providing valuable criticisms about the comparison of the two methods. Before replying in details to the revisions below, we would like to express a general point about our study design.

It is true that the methods that we compare in the central argument of the paper (Table 1) are inherently different. The benchmark represents a conventional breeding program in its advanced selection phase, and it does so in size, methods, and locations for what concerns breeding in Ethiopia (as detailed in the revised results section). 3D-breeding is an entirely different approach, never employed before, and is based on a different statistical model as much as it is based on a different interpretation of the breeding effort. Regardless of their differences, the two models are both aimed at providing recommendations to breeding effort and do that at the top of their potential. The comparison central to this paper is therefore aimed at their recommendation outcome, to see whether one method performs better than the other and why, while acknowledging their ultimate differences.

One must consider that the tricot design on which the 3D-breeding is based aims to maximize information gain from a sparse, unreplicated design. If in a hypothetical world we could measure everything that is routinely measured in centralized stations on thousands of farms, then it would be possible to compare methods in an orthogonal way. But this is a completely theoretical question, as it is not feasible to measure all correlated traits on the number of sites presented in this paper. The 3D-breeding tries to address exactly this situation and uses a ranking approach to address the limitations of data availability. This makes it a different approach from conventional genomic selection, and makes it more difficult to compare the two methods.

For this reason, already in the earlier version of the manuscript we made choices to enable the fair comparison of the two methods: i) we selected the same agroecologies for centralized and decentralized fields, ii) we derived accuracies in the same metrics (the Kendall Tau), iii)

we used a cross validation approach with seasons as blocks for both methods, iii) we used overlapping training and testing sets of genotypes in the benchmark as well as in the 3D-breeding.

In the revised version of the manuscript, we included new analyses to allow the reader to better compare the two methods in a fully quantitative way. At the same time, we carefully revised the text in the effort of making a point about the fairness of the comparison central to the paper.

Major comments:

1- Lines 292-296: On Dataverse (file: 4_DataFile_DecentralizedBreeding_DurumWheat.xlsx), there are 71 genotypes plus the 3 cultivars, please clarify this point. There were 41 genotype names in Figure S14 including the three recommended genotypes for the mega-environment (Asassa: 341ET_D; Hitosa: 390ET_D; Ude: 344ET_D).

Indeed, the full potential set of genotypes in decentralized fields was of 71, but we could not use it entirely because some genotypes were lacking genetic and phenotypic data in centralized stations. In the previous version of the manuscript, we opted to submit all the raw data and to filter the data in the scripts to get to the 41 eventually used. However, to acknowledge the reviewer point and to improve clarity and uniformity of the data and methods, we now filtered the input data to match the genotypes that we report in the manuscript.

This seems to be different from what described in the paragraph in which I understood that there were 41 genotypes plus 3 cultivars. Please reword.

It is indeed 38 traditional varieties plus 3 cultivars. This is now clarified in text (L131):

“A total of 1,165 decentralized field, each with 4 plots, were established between 2013 and 2015 during three growing seasons across the regions of Amhara (471), Oromia (399) and Tigray (295) (Fig. S1) using a subset of 38 purified landraces accessions identified through farmer evaluation in centralized trials²¹ and three modern cultivars, for a total of 41 wheat genotypes (Fig. S1).”

2- For each prediction scenario, the reference and the validation populations and their sizes were not always fully described. For example, in lines 342 to 349, it is not clear if the authors used the subset of the 41 genotypes in the reference or not. I would recommend adding a supplementary table describing different scenarios with the sizes of their training and validation populations.

Indeed, the subset of the 41 genotypes was included in the training population for the benchmark. To clarify this and the points below, we added a supplementary table (Table S3) breaking down the training and test information for every prediction scenario evaluated in the benchmark. We also clarified the materials and methods section as follows (L204):

“The benchmark considered two main prediction scenarios. In the first scenario, prediction was restricted to the centralized experiment. In this scenario, the genomic prediction model was trained on $GY_{STATION}$ and $OA_{STATION}$ measured on the full set of 400 genotypes evaluated in 2012, and the training dataset was $GY_{STATION}$ measured in the same locations in 2013 on the subset of 41 genotypes that were also included in the 3D-breeding. In the second scenario, the benchmark was trained on combined $GY_{STATION}$ and $OA_{STATION}$ data in centralized trials and used to predict the test population of 41 genotypes measured in decentralized fields for GY_{FARM} and OA_{FARM} . Mirroring the approach used in the 3D-breeding, the accuracy of genomic prediction in the second scenario was derived from a cross-validation approach averaging Kendall τ specific for Season 1, Season 2, and Season 3 using the square root of the sample size as weights³².

The benchmark was tested with additional prediction scenarios considering different training and test populations, including: i) without overlap between training and test samples, ii) restricting the training to the subset of 41 genotypes selected for 3D-breeding, iii) predicting GY_{FARM} and OA_{FARM} in decentralized fields stratified by their environmental distance from centralized stations.”

3- The reference population for the centralized trials consists of 400 genotypes scored in two seasons in two stations (expecting a total of 1,600 datapoints) while the top 41 genotypes of

them were selected for the decentralized trials with a total of 3,252 datapoints (inferred from the file: 4_DataFile_DecentralizedBreeding_DurumWheat.xlsx; for these 41 genotypes).

The revised version of the manuscript includes several clarifications about this point. Indeed, the total number of datapoints is not markedly different in the two approaches: centralized trials consist of replicated plots for each of the 400 genotypes, meaning 2 locations x 2 seasons x 2 reps x 400 genotypes = 3,200 datapoints. We consider decentralized farms to provide 4,660 datapoints (4 plots in each of 1165 farms). The size of the population is now clarified throughout the text and in the Statistics and Reproducibility section at the end of the Materials and Methods (L277):

“Centralized experiments were run in two locations, for two seasons, on replicated plots for 400 genotypes for a total of 3,200 plots. The benchmark was run with different prediction scenarios considering separated and overlapping training and test populations as specified in the methods. Decentralized trials were performed on 1,165 farmer fields, with four plots per farmer field evaluated in ranking, for a total of 4,660 plots.”

Still, there are caveats in interpreting side by side the number data points collected in centralized stations and in decentralized farms, as farm data relies on ranking and not on measurements. In the revised version of the paper, we worked to show that the different performance of the two methods does not depend on the amount of datapoints, rather in their information content. To this end, we included a new analysis to allow a comparison of the two methods when the number of observations per genotype is similar. We run five additional 3D-breeding scenarios, each of which considering a subsample of the full dataset comprised of 4,660 observations. These subsets are run considering a different number of observations per genotype, from 75% (amounting to 85 plots per genotype) to 5% (amounting to 5 plots per genotype). The latter scenario matches the number of observations collected in centralized fields, that amount to 8 plots per genotype. We run these scenarios with 100 random subsamples each, finding that 3D-breeding still outperforms the benchmark. The new analysis is discussed in the “Influence of data volume on the comparison between 3D-breeding vs centralized breeding” paragraph in the supplementary methods:

“The comparison between the two approaches should be fair. We consider that the two setups are comparable in terms of the costs and efforts but this is difficult to quantify for a situation in which each of these approaches would be used on a routine basis. One key difference, however, is the volume of data available to each, and this is also a key cost driver. Centralized breeding involved 8 plots per genotype (two locations, two years, two replications per location), while decentralized observations were conducted on 113 plots per genotype in the tricot incomplete block design. So, 3D-breeding has many more datapoints per genotype. To assess the sensitivity of 3D-breeding to the data volume for its prediction accuracy, we created 5 scenarios that represent different data volumes available. Each scenario is a subset of the plots containing respectively 75% (85 plots), 50% (56 plots), 25% (28 plots), 15% (17 plots) and 5% (5 plots) of the data. We selected plots randomly, keeping balance between seasons, and always including all 41 genotypes evaluated. For each prediction we calculated the Kendall τ correlation with the observed data. This process was repeated 100 times per scenario and averaged. We report the average Kendall τ correlation for the different scenarios in Table S4.”

Results are reported in Table S4 and in the results section (L335):

“Model predictions from 3D-breeding consistently provided higher accuracy than the benchmark for GY_{FARM} and OA_{FARM} , with $\tau = 0.109$ and $\tau = 0.251$ (Table 1). When supported by smaller sets of observations (from 5% to 75% of the available data), 3D-breeding maintained superior accuracy than the benchmark, with a mean accuracy spanning from $\tau = 0.162$ to $\tau = 0.230$ for OA_{FARM} and from $\tau = 0.076$ to $\tau = 0.106$ for GY_{FARM} (Table S4).”

Thus, there is no wonder why 3D-Breeding performed better than the centralized breeding in your experiment given that:

a) Your decentralized reference population is exactly the same as the validation population (41 genotypes) compared to the centralized populations that involved 400 different genotypes

It is true that the decentralized reference population is the same as the validation population. 3D-breeding, based on tricot design, entails a complex network of partial rankings to reconstruct a general ranking of genotypes. In this highly interconnected design, it is not

possible to remove a training set of genotypes, as this would collapse the rankings network and its efficacy. We are aware that this condition may inflate accuracy, and for this reason we run the benchmark with overlapping training and test samples. This provides the benchmark an advantage and makes it more competitive. In the revised version, we computed further prediction scenarios in which we used different combinations of train and test sets (Table S3), predicting the 41 decentralized genotypes with i) all station data (without cross-validation), ii) non-overlapping station data, and iii) the 41 genotypes measured in stations. None of these alternative scenarios could consistently surpass accuracy of the 3D-breeding for both OA and GY. This finding reinforces the value of combining the two methods to improve varietal recommendation and breeding. Implications are discussed in the paragraph “Implications for rethinking breeding programs” discussed below in more detail.

b) The size of the decentralized reference population seems to be larger than the size of the centralized reference population I acknowledge that it will be impossible to convince the farmers to grow the remaining 359 genotypes with lower performance in the centralized trials, but another design should have been carried out to better synchronize both comparisons. At least similar reference sizes should have been used if not already done. Another possible way is to use a subset of the station genotypes (other than selected 41 genotypes) as a validation population to compare predictability of the station and the farm references.

The tricot approach design used in 3D breeding considers an on-farm trial as an incomplete block of four varieties. The data from a given season is collated into a single dataset which constitute the trial. The trial is the combination of all data of farmers coming together. Since varieties co-occur in the same network, the Plackett Luce can collapse the data in a ranking reasoning. Hence, it is not straightforward to evaluate side by side the methods when it comes to sizes of the reference populations. In the revised version of the paper, we added two new analyses to better synchronize both comparisons: the first looking at different prediction scenarios of the benchmark (Table S3), the second looking at different prediction scenarios of the 3D-breeding (Table S4). Combining the two, we report that the accuracy of the decentralized data was still superior. This is detailed in the responses above. We address this point in a paragraph in the discussion section entitled “Implications for rethinking breeding programs” (L373):

“Our results show that 3D-breeding is superior to a benchmark that represents a centralized breeding approach. The genomic prediction benchmark and 3D-breeding rely on different statistical designs and methods, yet they have the same aim: providing accurate prediction of phenotypes in untested environments. We believe that the implementation of the two approaches was realistic and of high quality, making the comparison realistic. We have explored whether the superiority of 3D-breeding is sensitive to the influence of data availability, the geographical placing of the centralized selection environments or the variable of focus (overall appreciation or grain yield) and found that its superiority is robust. This has important implications for breeding program design.”

Still, we acknowledge the reviewer’s point in the Discussion section (L455)

“In this study, farmers evaluated top performing varieties chosen from a larger set, but future studies may focus on larger collections of germplasm to be evaluated through 3D-breeding in combination with evaluations performed in research stations. These may include new genetic materials prioritized by speed-breeding⁷⁶ and haplotype-based selection⁷⁷. Our results show that already the current replication level of the experimental design may support more diversity (Table S4).”

4- Looking at Figure S14, the best performing genotypes for the three nodes are 341ET_D, 344ET_D, 390ET_D, 441ET_D and 033ET_D. The first three of these are the three recommended genotypes for the mega-environment (Asassa, Ude and Hitosa; as described in the file: 4_DataFile_DecentralizedBreeding_DurumWheat.xlsx). This makes the results of Figure 2A very odd.

Thank you for pointing that out. There was a mistake in ordering the labels for the plot: we were implementing a method to plot the coefficients with error bars and the decision tree and mistakenly added the labels in the wrong direction. In the revised manuscript, we fixed the plot to match with the model coefficients. We wish to point out that the previous (mistaken) figure was just a display chart, and it did not affect any of the conclusions of the research and/or the model predictions reported in the main text.

Moreover, in the same file, Asassa and Hitosa showed the lowest average “farmer_rank” with 1.74 and 1.67; respectively, which is against what is described in lines 155 to 157.

This confusion stems from the fact that the Plackett Luce approach is not based on value averages, but rather on a combination of partial rankings. Taking the mean of “farmer_rank” is not informative and cannot be compared with the overall results because of the non-replicated nature of crowdsourcing plots. Indeed, in Plackett Luce, the probability of selecting one genotype is relative to the overall rankings and not on individual observations. This follows the Luce’s axiom of choice (Luce 1959, 1977), that is now clarified in the methods section (L166):

“For the analysis of the decentralized data, we used the Plackett-Luce model^{26,27}, implemented in the R package PlackettLuce²⁸. The use of the Plackett-Luce model to analyze data from decentralized crop variety trials is demonstrated by van Etten et al. 2019¹⁶. Plackett-Luce is a rank-based model that follows the Luce’s axiom of choice²⁷, which assumes that ranking order between every pair of options does not depend on the presence or absence of other options. The model estimates the *worth* parameter α which is related to the probability (P) that one genotype i wins against all other n genotypes in set, and is obtained using the following equation:”

5- I would love to see a prediction scenario in which both centralized and decentralized data were used together in the training population, this may further have higher prediction accuracy

This is an interesting perspective and indeed we are hoping to implement this scenario in future works. Here we have limited data that we cannot fully exploit for a similar comparison, so we prefer to focus on the central comparison between the two methods to validate 3D breeding as a mean to gain information. We add the approach suggested by the reviewer in the conclusion section of the manuscript (L490):

“Ideally, 3D-breeding could be combined with conventional, centralized breeding to improve the training of prediction models to address local adaptation. Once new varieties are developed through the crowdsourced combination of breeders’ and farmers’ knowledge, future research shall focus on the potential impact of these methods on conservation and use of

traditional agrobiodiversity both *in situ* and beyond the local environments in which it was developed.”

Minor comments:

Line 24: I guess you have 1,165 farms in each one you have four plots. Please reword throughout the manuscript to avoid confusion

This is correct. We clarified this point throughout the manuscript.

I would prefer to calculate heritability and genetic correlation with GREML method, e.g. using GCTA or MTG2 software.

To address this request, we added a new model calculating heritability by including genetic covariance. For consistency with the rest of the analyses, this was accomplished by linking the relationship matrix to regressor variables with R-ASReml. The new h^2 computation corresponds to narrow sense heritability (as opposed to broad sense heritability), and it was included in supplemental table S1 and S2 and discussed in text (L292):

“Heritability (H^2), the proportion of phenotypic variance explained by genotypic variance, was 0.55 and 0.42 for $GY_{STATION}$ across locations for 2012 and 2013 respectively (Table S1). To capture farmers’ traditional knowledge regardless of gender, farmer scores were combined across men and women respondents. The H^2 of $OA_{STATION}$ was 0.78 across locations. Narrow sense heritability (h^2) was calculated considering genetic co-variance of genotypes and provided a more conservative estimate for all traits, yet $OA_{STATION}$ was consistently more heritable than $GY_{STATION}$ (Table S1, Table S2).”

Table S2: The heritability of OA is very high. I assume that the farmer’s selection in Ethiopia will be highly affected by heading date. Appreciate if you can check the correlation between OA and other recorded traits, especially phenology related traits.

Indeed, it is. The correlation of OA with other traits was extensively explored in Mancini et al 2017 and Kidane et al 2017. Although earliness is not the only trait that contributes to OA,

farmers are very aware of flowering time and consistently prefer early flowering genotypes. Farmers' preference is correlated with yield, seed size, biomass. This is now clarified in the results (L309):

“Previous studies explored the relation between OA and agronomic performance of wheat, showing that farmers' appreciation was positively correlated to yield, seed size, biomass, and negatively correlated with time to flowering and time to maturity^{20,21}.”

Is there any previous study on the level of the within genetic diversity for the studied landraces? And how many seeds you used for 90K genotyping per landrace?

Very often, landraces are cultivated as heterogenous materials and this is reflected in the accessions stored in seedbanks. When we withdrew landraces from the Ethiopian Biodiversity Institute, we performed purification fields to derive genetically homogenous lines to be used in all experiments downstream. The original accessions were multiplied spike-to-row for two seasons as described in Mengistu et al 2016: all seeds that were genotyped and were grown in centralized stations and decentralized farms all derive from a single spike purified from the original accession. Genotyping was performed on a pool of 5 seedlings per accession. This is now clarified in the materials and methods (L89):

“We selected 400 durum wheat (*Triticum durum* Desf.) genotypes from a representative collection of landrace accessions maintained at the Ethiopian Biodiversity Institute (EBI) and improved lines cultivated in Ethiopia. Landrace accessions were purified to derive a uniform genetic background to undergo all subsequent analyses: all seeds used in this study derived from a single spike representative of the EBI accession as described in Mengistu et al. (2016)¹⁹.”

If the authors have information about the age of the farmers, I think it worth to do a correlation based on the age other than the gender as this will reflect the experience of the farmer.

We agree that this would be an interesting perspective. However, in the current dataset, we are not able to stratify for gender or age (see response to Reviewer#2). This is mentioned in the conclusion section of the manuscript (L489):

“Further 3D-breeding studies may opt to stratify participants for socioeconomic features of interest, including gender, age, or income, to fully characterize traditional knowledge in its many dimensions. Ideally, 3D-breeding could be combined with conventional, centralized breeding to improve the training of prediction models to address local adaptation.”

Reviewers' Comments:

Reviewer #2:

Remarks to the Author:

The authors adequately addressed the issues I raised.

Reviewer #3:

Remarks to the Author:

This is one of the very few papers I really enjoyed reading and was waiting to get the author's response to my comments, which I found very satisfactory. Table S3 showed that using a reference of selected genotypes in stations had comparable accuracy with 3D-breeding for GY ONLY. However, the standard deviation was more than double that in 3D-breeding which again proves its superiority. I would love to see the paper published in the current form and I cannot wait to apply it in real life breeding programs.

Abdulqader Jighly